DOI: 10.1038/s41467-017-01592-3　　**OPEN**

# A multiplexable TALE-based binary expression system for in vivo cellular interaction studies

Markus Toegel 🆔 [1], Ghows Azzam[1,3], Eunice Y. Lee[1,4], David J.H.F. Knapp[1], Ying Tan[2], Ming Fa[2] & Tudor A. Fulga[1]

Binary expression systems have revolutionised genetic research by enabling delivery of loss-of-function and gain-of-function transgenes with precise spatial-temporal resolution in vivo. However, at present, each existing platform relies on a defined exogenous transcription activator capable of binding a unique recognition sequence. Consequently, none of these technologies alone can be used to simultaneously target different tissues or cell types in the same organism. Here, we report a modular system based on programmable transcription activator-like effector (TALE) proteins, which enables parallel expression of multiple transgenes in spatially distinct tissues in vivo. Using endogenous enhancers coupled to TALE drivers, we demonstrate multiplexed orthogonal activation of several transgenes carrying cognate variable activating sequences (VAS) in distinct neighbouring cell types of the *Drosophila* central nervous system. Since the number of combinatorial TALE–VAS pairs is virtually unlimited, this platform provides an experimental framework for highly complex genetic manipulation studies in vivo.

[1] Weatherall Institute of Molecular Medicine, Radcliffe Department of Medicine, University of Oxford, Oxford OX3 9DS, UK. [2] GenetiVision Corporation, 8874 Interchange Drive, Houston, TX 77054, USA. [3] Present address: School of Biological Sciences, Universiti Sains Malaysia, Penang 11800, Malaysia. [4] Present address: Columbia University, College of Physicians and Surgeons, New York, NY 10032, USA. Markus Toegel and Ghows Azzam contributed equally to this work. Correspondence and requests for materials should be addressed to T.A.F. (email: tudor.fulga@imm.ox.ac.uk)

Binary modular expression systems have revolutionised our ability to manipulate and dissect regulatory networks in vivo. Foremost, the yeast-derived GAL4–UAS system has become an invaluable tool for disentangling intricate genetic interactions in *Drosophila*[1], mice[2], and zebrafish[3]. However, the extraordinary complexity of many biological contexts, such as the nervous system, requires the design of more sophisticated tools allowing for parallel and controlled manipulation of multiple tissues and cell types. For example, the field of connectomics, dedicated to deciphering neuronal circuits, would greatly benefit from the ability to simultaneously label different populations of

neurons and to control their excitatory state (activation or inhibition) independent of each other. To enable such studies, two other binary expression systems have been developed in *Drosophila*, based on either the viral DNA binding protein LexA, which recognises the LexA operator[4], or the fungal DNA binding protein QF acting on the QUAS sequence[5,6]. Although valuable, each of these systems relies on a single exogenous DNA binding protein (driver) that acts on a unique recognition motif (responder), and thus cannot be used to independently express multiple transgenes in separate tissues. Consequently, the only present modality to carry out complex cellular interaction studies

**Fig. 1** Design and optimisation of TALE–VAS driver–responder pairs in *Drosophila* S2 cells. **a** Schematic representation of the TALE–VAS system. TALEs consist of a N-terminal domain (TAL N′) containing a nuclear localisation sequence (NLS), a custom TALE-array, and the C-terminal domain (TAL C′) fused to a transcription activation domain (VP64). Each repeat within the TALE-array recognises a specific nucleotide depending on the amino acids 12 and 13 of that repeat (NG = T, NI = A, NN = G, HD = C). Transcription of the gene of interest is initiated upon binding of the TALE to its cognate 20 nt long VAS. The invariable 5′ T at position 0 is shown in blue. **b**, **c** Comparative analysis of reporter expression between four engineered TALE–VAS driver–responder pairs. Diagrams indicate the experimental conditions tested, reflecting the drivers used (circles) and their respective responders (shapes with matching indentations)—colours and numbers indicate the four different TALE–VAS pairs (**b**). **c** Flow cytometry analysis of reporter expression in *Drosophila* S2 pMT-GAL4 cells. The original backbone TALE plasmid without a TALE-array (TALE_CTR) was used to establish background levels of GFP expression. **d**, **e** Specificity of transgene activation by TALE drivers. Each VAS responder was co-transfected with TALE_{1, 3, 4} drivers and the control TALE in S2 pMT-GAL4 cells (**d**). **e** Flow cytometry analysis of EGFP reporter induction for each TALE–VAS combination in (**d**). **f**, **g** Comparison of TALE–VAS and GAL4–UAS systems. S2 pMT-GAL4 cells were transfected in parallel with either TALE_4 > VAS_4-EGFP or GAL4 > UAS-EGFP driver–responder pairs (**f**). Efficiency of transgene expression assessed by flow cytometry and confocal microscopy (scale bar = 5 μm) (**g**). In all cases the mean EGFP fluorescence was calculated from three biological replicates (n = 3 from one experiment, mean ± s.d.; a.u. arbitrary units)

is to combine different expression systems. Such experiments, however, have so far been restricted to parallel analyses involving maximum two platforms, such as GAL4 and LexA[7] or GAL4 and QF[5]. Furthermore, to increase the number of consecutive manipulations, more such orthogonal driver–responder pairs would have to be adopted from different organisms and engineered to render them compatible with in vivo applications and to avoid cross-platform interactions.

To address this limitation and simplify the range of tools, we have developed a customisable binary expression system based on bacterial transcription activator-like effector (TALE) fusion proteins. We reasoned that since TALEs can be programmed to bind any DNA sequence, this platform could enable the generation of virtually unlimited driver–responder pairs suitable for independent parallel transgene expression in vivo. By analogy to the GAL4 upstream activating sequence (UAS), we have termed the programmable TALE responder binding sites variable activating sequences (VAS). Using this logic, we demonstrate that orthogonal TALE drivers and VAS responders can be combined in vivo to enable simultaneous expression of multiple transgenes in interacting tissues, without any apparent crosstalk between individual pairs. Furthermore, we show that the TALE–VAS system can be used in conjunction with other binary expression platforms, further expanding the promise of this technology for complex cellular interactions studies.

## Results

**Design and optimisation of TALE–VAS pairs in vitro.** TALEs were discovered in the bacterial plant pathogen *Xanthomonas* where they act as secreted effector proteins that induce host gene expression[8,9]. DNA binding is achieved through arrays of 34 amino acid long repeats each containing a repeat-variable diresidue (RVD) at position 12 and 13, which provide the code for nucleotide (nt) recognition in the major groove of the DNA[10,11] (Fig. 1a). In order to assess the general feasibility of our approach, we first generated four different TALE drivers based on the pJC-TALE-VP64 vector[12], which is optimised for GAL4-mediated transgene expression in *Drosophila* cells (Supplementary Fig. 1). The four corresponding VAS responders (VAS$_1$-EGFP, VAS$_2$-EGFP, VAS$_3$-EGFP, VAS$_4$-EGFP) were created by cloning TALE target sequences upstream of an EGFP reporter construct in the *Drosophila* pJFRC81 vector scaffold[13] (Supplementary Fig. 2). To ensure high specificity and strong affinity, TALEs were designed to recognise 20 nt long VAS sequences.

Although all current RVDs have high affinity for specific nucleotides, some can tolerate less favourable interactions[10,11]. Consequently, it has been reported that under certain circumstances both natural and synthetic TALEs could bind to off-target sequences containing up to three base pair mismatches[10,11,14]. This is an important consideration for the use of TALEs as programmable nucleases in genome editing applications, where introduction of DNA double-strand breaks at off-target sites can result in unintended mutagenesis events[15,16]. In the TALE–VAS system confounding off-target effects could manifest in the ectopic activation of endogenous genes. However, the risk of such events is mitigated to some degree by the requirement of TALEs to bind in close proximity to a genomic promoter or enhancer. To identify putative VAS off-target sites, we carried out a comprehensive bioinformatics search for sequences containing up to four nt mismatches across a 4 kb window centred on the transcription start sites (TSS) of all annotated *Drosophila* genes. This analysis revealed that none of the four TALE drivers had perfect matching endogenous targets or off-target sites containing single-nucleotide mismatches (Supplementary Fig. 3a, b and Supplementary Data 1). Furthermore, only TALE$_2$ had a small

number of possible off-targets containing two and three nt mismatches. TALE$_1$, TALE$_3$, and TALE$_4$, however, displayed no relevant off targets with up to three nt mismatches, rendering them well suited for subsequent in vivo applications.

GAL4–UAS responder lines usually carry 5, 10, or 20 UAS copies translating into increasing strength of transgene activation[17]. To assess the requirements of the TALE–VAS system, we first used TALE$_1$ to drive EGFP expression downstream of 1, 5, or 10 VAS$_1$ repeats. These constructs were tested in *Drosophila* S2 cells harbouring a stably integrated Cu$^{2+}$-inducible metallothionein promoter GAL4 (pMT-GAL4), which was used to drive TALE expression. In contrast to the GAL4–UAS system, fluorescence levels quantified by flow cytometry already saturated at 5 repeats suggesting that strong transgene expression could be achieved even with a relatively low number of VAS copies (Supplementary Fig. 4). Since further analysis revealed that reducing the number of VAS repeats to 3 only marginally impacted reporter expression, all final responders were designed with 3 VAS copies to reduce the number of repetitive sequences.

To establish the efficiency of transgene expression across different combinations of drivers and responders, we then tested each TALE–VAS pair in S2 pMT-GAL4 cells (Fig. 1b, c and Supplementary Fig. 5). In addition to lacking relevant off-targets (Supplementary Fig. 3a, b), TALE$_1$, TALE$_3$, and TALE$_4$ showed the strongest EGFP reporter activation and thus were used in all subsequent experiments. To test the orthogonality of TALE–VAS pairs we co-transfected each TALE plasmid with either cognate or non-cognate VAS responder plasmids. In all cases, robust EGFP activation above background levels was only observed when the cognate responder was present (Fig. 1d, e and Supplementary Fig. 6). To benchmark the TALE–VAS constructs against the established GAL4–UAS system, we directly compared the performance of these platforms in the same assay. Since the respective DNA binding proteins have different rates of synthesis, folding, maturation, and degradation, this assay could only provide a qualitative rather than quantitative readout. To minimise experimental interference, we used the same setup in pMT-GAL4 S2 cells by swapping the TALE$_4$–VAS$_4$ responder pair for a GAL4 plasmid and corresponding UAS responder. Under these experimental conditions, the TALE–VAS system rendered comparable transgene activation with the GAL4–UAS system (Fig. 1f, g). Finally, to establish basal responder activity in pMT-GAL4 S2 cells, we measured VAS$_4$-EGFP and UAS-EGFP expression in the absence of any driver and CuSO$_4$. In contrast to the strong activation observed in the presence of the corresponding TALE$_4$ or GAL4 drivers, both responders alone displayed low background signal above the untransfected control cells (Supplementary Fig. 7).

**TALE-mediated activation of VAS-responders in vivo.** Having established the viability of driving orthogonal transgene expression by the TALE–VAS system in vitro, we next sought to implement this technology in vivo by creating transgenic driver–responder lines in *Drosophila*. To achieve tissue specificity for targeted transgene expression, we cloned different enhancer elements upstream of the hsp70 minimal promoter in the TALE$_1$, TALE$_3$, and TALE$_4$ vectors. To demonstrate the versatility of this system, we focused on enhancers that have distinct but also broad expression patterns in neighbouring tissues and can be analysed in third larval instars, which are readily amenable to dissections and imaging. To this end, we generated TALE drivers for mature neurons (elav$^{1.8kb}$-TALE$_1$), neuroblasts (ase$^{0.8kb}$-TALE$_3$), muscle cells (mhc$^{2.4kb}$-TALE$_3$), and glial cells (repo$^{1.9kb}$-TALE$_4$) (Fig. 2a, Supplementary Fig. 8, and Methods). Additionally, we generated a new set of responder constructs expressing three tagged

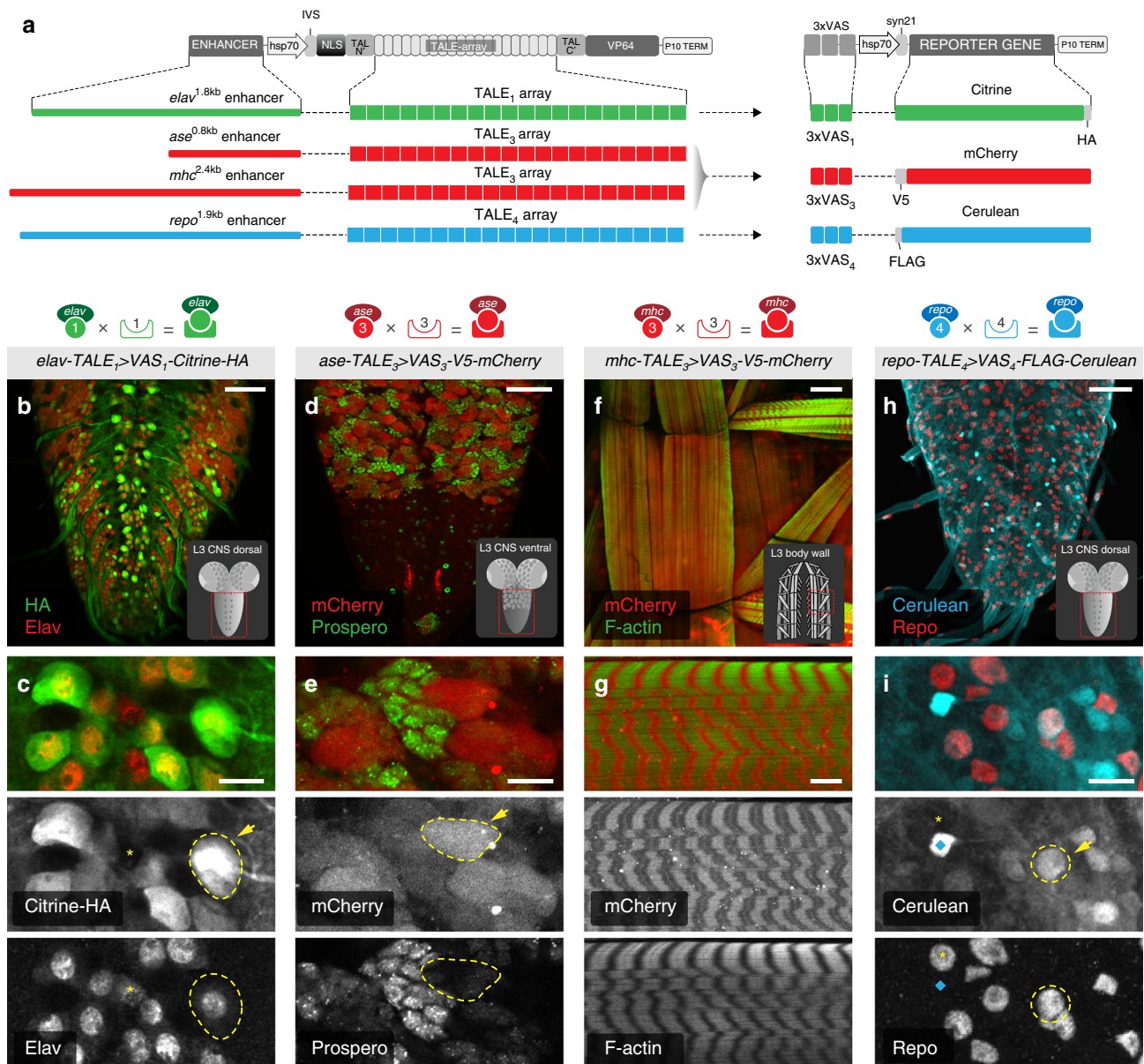

**Fig. 2** Implementation of TALE–VAS binary expression systems in vivo. **a** Schematic representation of the TALE–VAS constructs used in flies. The $elav^{1.8kb}$, $ase^{0.8kb}$, $mhc^{2.4kb}$, and $repo^{1.9kb}$ enhancers were cloned upstream of $TALE_{1, 3, 4}$, and the corresponding 3× VAS sequences were placed in front of Citrine, mCherry, and Cerulean. **b–i** TALE–VAS-based targeted gene expression in single tissues. Insets show imaging location within each target tissue (red frame). **b** Dorsal view of $elav^{1.8kb}$-$TALE_1 > VAS_1$-Citrine-HA third instar larval ventral nerve cord (VNC) stained with anti-HA (transgene expression) and anti-Elav antibody (neuronal marker). **c** Close up illustrating the overlap between Citrine-HA and nuclear Elav staining in neurons (yellow dashed line and arrow). Some Elav-positive cells lack Citrine-HA expression (yellow asterisk). **d** Ventral view of $ase^{0.8kb}$-$TALE_3 > VAS_3$-V5-mCherry third instar larval VNC showing mCherry expression in neuroblasts (red) and Prospero staining of ganglion mother cells (anti-Pros antibody, green). **e** Detailed, high magnification of a single neuroblast (yellow dashed outline and arrow) and surrounding ganglion mother cells (green). **f** Third instar larval body wall musculature from $mhc^{2.4kb}$-$TALE_3 > VAS_3$-V5-mCherry at low magnification. **g** High-magnification sarcomere morphology showing alternating bands of mCherry expression and F-actin characteristic of striated muscles. **h** $repo^{1.9kb}$-$TALE_4 > VAS_4$FLAG-Cerulean third instar larval VNC stained with an anti-GFP antibody (transgene expression) and anti-Repo (glia marker). **i** Detailed view of subperineural glia showing cells that stain positive for both Cerulean and Repo (yellow dashed outline and arrow) express only Cerulean (blue diamond) or only Repo (yellow asterisk). Scale bars = 50 μm (**b**, **d**, **f**, **h**), 10 μm (**c**, **e**, **g**, **i**)

fluorophores under the control of corresponding VAS elements ($VAS_1$-Citrine-HA, $VAS_3$-V5-mCherry, and $VAS_4$-FLAG-Cerulean) (Fig. 2a). To reduce the number of responder constructs and facilitate subsequent in vivo applications of this technology, we combined the $VAS_1$ and $VAS_4$ responders in a single vector. All TALE drivers and VAS responders were integrated into the *Drosophila* genome by PhiC31 integrase-mediated germline

transformation using five different landing sites (Supplementary Fig. 9). For each construct, at least two independent transgenic lines were obtained that did not show any obvious differences in their expression pattern and thus were used interchangeably.

Whole mount analysis of the two responder lines ($VAS_3$-V5-mCherry and $VAS_1$-Citrine-HA, $VAS_4$-FLAG-Cerulean) revealed weak background expression in a small number of tissues

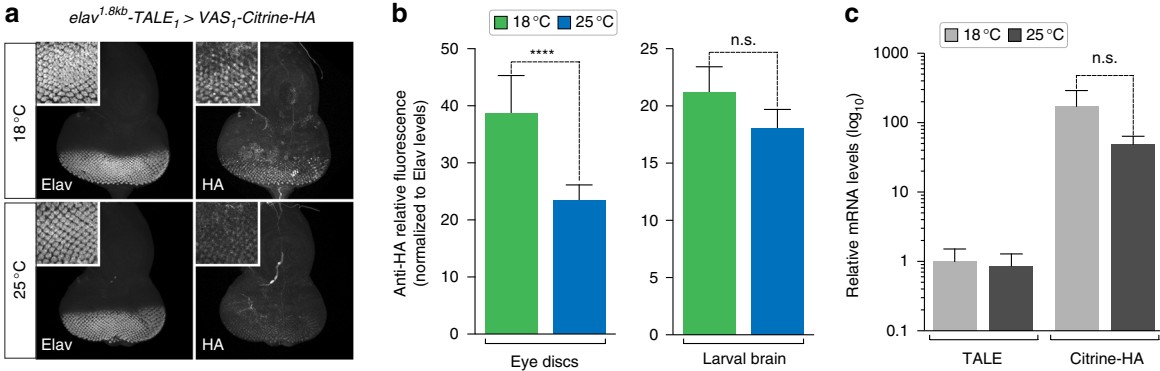

**Fig. 3** Impact of temperature on TALE–VAS activity. **a** Eye discs from $elav^{1.8kb}$-$TALE_1$ > $VAS_1$-Citrine-HA larvae stained with anti-Elav and anti-HA (transgene expression) antibodies reared at 18 and 25 °C. Insets show detail of the retina used to quantify fluorescence levels. Scale bar = 100 μm. **b** Quantification of Citrine-HA protein levels (anti-HA fluorescence normalised to anti-Elav levels) in eye discs and larval brains at 18 °C (n = 11 discs; n = 3 brains) and 25 °C (n = 10 discs; n = 4 brains; mean ± s.d. unpaired t-test n.s. P > 0.05, ****P < 0.0001). **c** RT-qPCR quantification of $TALE_1$ and Citrine-HA mRNA expression from dissected third larval instar brains at 18 and 25 °C (n = 3 biological replicates, 10 brains each; mean ± s.d. unpaired t-test n.s. P > 0.05)

(Supplementary Fig. 10a, b). Since this pattern is highly restricted and reproducible, it may reflect a positional integration effect or the presence of putative transcription factor binding sites within the VAS sequence. Ectopic expression was also observed in the epidermis when VAS transgenes were driven by the $mhc^{2.4kb}$, $ase^{0.8kb}$, or $elav^{1.8kb}$ TALE driver (Supplementary Fig. 10c). Because the $mhc^{2.4kb}$-$TALE_3$ and the $ase^{0.8kb}$-$TALE_3$ are both integrated at the same landing site (VK37), it is conceivable that this additional expression also results from TALEs activation by endogenous regulatory elements associated with the integration site. Although the ectopic expression observed using these driver–responder pairs is unlikely to interfere with functional studies, it could under certain circumstances compromise the interpretation of experiments employing effector transgenes. The weak VAS background may also in some instances impact the ability to establish and maintain responder lines encoding lethal transgenes. If necessary, these issues could be addressed by screening the large VAS sequence space for variants devoid of any baseline activity and choosing alternative or random genomic integrations sites for establishing new TALE drivers and VAS responder lines. These considerations will be particularly important if attempting to generate TALE–VAS libraries, in which case the specificity of all system components should be rigorously assessed and empirically validated across multiple tissues and developmental stages.

To assess the levels of transgene activation and their pattern of expression, each TALE driver was crossed individually to its cognate responder line and third larval instar brains and body wall muscles were analysed by immunofluorescence and antibody staining. In all cases, we observed strong tissue-specific activation in a pattern predominantly overlapping with the expression of the *mhc*, *elav*, *ase*, and *repo* genes, respectively (Fig. 2b–i). It should be noted that although these enhancers were selected to best reflect the endogenous expression patterns of their associated genes, it is unlikely that the chosen sequence boundaries captured the entire set of native regulatory elements. Consequently, only a partial overlap should be expected between the TALE-drivers expression domains and those of the corresponding endogenous genes (antibody staining).

To confirm that the TALE drivers do not cross-activate non-cognate responders in flies, we generated a triple VAS responder line ($VAS_1$-Citrine-HA, $VAS_3$-V5-mCherry, $VAS_4$-FLAG-Cerulean) by classic recombination, and crossed it to each TALE driver line. Similar to the results in S2 cells, expression was only

observed between matching TALE–VAS pairs and no unspecific activation of non-cognate responders was detected in the target tissues, demonstrating the specificity of this system in vivo (Supplementary Fig. 11).

The GAL4–UAS system has been frequently used at different temperatures to either enhance or attenuate transgene expression levels[18]. To assess whether the TALE–VAS system also displays temperature sensitivity, we analysed reporter levels by anti-HA antibody staining in brain and eye disc tissues of $elav^{1.8kb}$-$TALE_1$ > $VAS_1$-Citrine-HA third larval instars from crosses reared at 18 or 25 °C. Although expression was robust at both temperatures, Citrine-HA protein levels appeared higher at 18 °C relative to 25 °C, which is opposite to the GAL4–UAS system behaviour (Fig. 3a, b). The amplitude of this effect appeared to vary between tissues, and only reached statistical significance in the eye discs. To determine if this tendency is attributed to a temperature-induced change in TALE production or in the potency of TALE-VP64 to drive VAS transgene expression, we measured TALE and Citrine transcript levels by RT-qPCR. This analysis revealed that while the Citrine mRNA levels displayed an increase trend at 18 °C compared to 25 °C, TALE expression remained largely unaffected by temperature (Fig. 3c). Together, these results suggest that TALE-VP64 activity is slightly stabilised or enhanced at lower temperatures.

**Multiplexed TALE–VAS transgene expression in the nervous system**. Although the TALE–VAS system is a self-sufficient platform for multiplex transgene expression, the ability to combine it with other binary expression systems could be beneficial under certain circumstances. For example, using TALE–VAS driver–responder pairs in parallel with the extensive repertoire of GAL4–UAS constructs could enable genetic and cellular interaction studies of unprecedented complexity. To test this possibility, we first crossed the $ase^{0.8kb}$-$TALE_3$; $VAS_3$-V5-mCherry line to the UAS-mCD8-GFP; ase-GAL4 line to directly compare the two systems in the same tissue. Analysis of larval central nervous system (CNS) from progeny animals revealed simultaneous and comparable expression of both transgenes. Although the two drivers harbour different *ase* enhancer elements (see Methods) and the two transgenes are localised to distinct cellular compartments (cytoplasm and membrane), their pattern of expression in neuroblast cells was predominantly overlapping (Fig. 4a and Supplementary Fig. 12). To assess the orthogonality of these systems, we then generated a triple TALE driver line ($elav^{1.8kb}$-

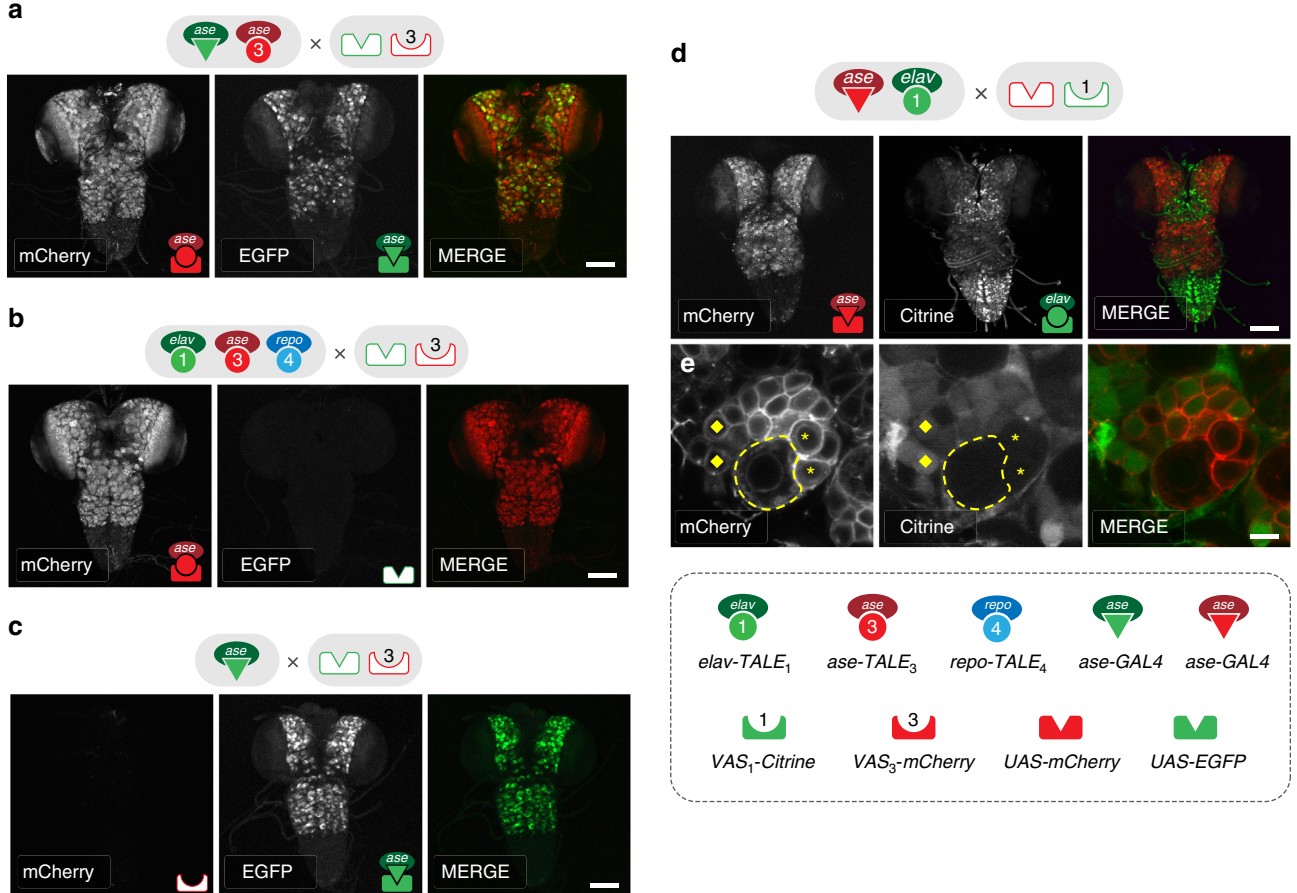

**Fig. 4** Parallel delivery of TALE–VAS and GAL4–UAS binary expression systems in vivo. **a** Analysis of mCherry and GFP expression in larval CNS neuroblast cells from $ase^{0.8kb}$-$TALE_3$; ase-GAL4 > $VAS_3$-V5-mCherry; UAS-mCD8-GFP animals. **b**, **c** Orthogonal transgene activation by TALE–VAS and GAL4–UAS systems. $ase^{0.8kb}$-$TALE_3$ specifically activates $VAS_3$-V5-mCherry in neuroblasts without driving expression of the UAS-mCD8-GFP transgene (**b**). Conversely, ase-GAL4 activated only the UAS-controlled mCD8-GFP transgene (**c**). **d** Simultaneous labelling of the neuroblast (mCherry) and neuronal (Citrine) compartments by combinatorial $elav^{1.8kb}$-$TALE_1$; ase-GAL4 > $VAS_1$-Citrine-HA; UAS--mCD8-mCherry transgene expression. **e** Detail of a large neuroblast (dashed line) devoid of Citrine, surrounded by ganglion mother cells (asterisk) and differentiating neurons (diamond) gradually turning on Citrine expression. Scale bars = 100 μm (**a**–**d**), 5 μm (**e**)

$TALE_1$, $ase^{0.8kb}$-$TALE_3$, $repo^{1.9kb}$-$TALE_4$) and crossed it to the mixed $VAS_3$-V5-mCherry; UAS-mCD8-GFP responder line. While $ase^{0.8kb}$-$TALE_3$ drove robust mCherry expression in neuroblasts, none of the TALE drivers activated the UAS transgene to any detectable levels (Fig. 4b). Conversely, crossing the ase-GAL4 driver to the same mixed responder line resulted in specific activation of only the UAS-mCD8-GFP responder in neuroblasts, indicating that GAL4 drivers are also unable to interact with VAS responders (Fig. 4c). To demonstrate the utility of this approach in driving independent transgenes in interacting tissues, we then expressed one marker in neurons using the TALE–VAS system ($elav^{1.8kb}$-$TALE_1$ > $VAS_1$-Citrine-HA) and a second transgene in neuroblasts with GAL4–UAS (ase-GAL4 > UAS-mCD8-mCherry). In this instance, each driver specifically activated its cognate responder in the corresponding tissues with no crosstalk between the two platforms (Fig. 4d, e).

Next, we sought to establish the potential of this platform for driving parallel activation of multiple transgenes in several neighbouring tissues in vivo. To this end, we first crossed the triple TALE driver line ($elav^{1.8kb}$-$TALE_1$, $ase^{0.8kb}$-$TALE_3$, $repo^{1.9kb}$-$TALE_4$) to the triple VAS responder line ($VAS_1$-Citrine-HA, $VAS_3$-V5-mCherry, $VAS_4$-FLAG-Cerulean), thereby generating a viable animal expressing three distinct transgenes in

non-overlapping tissues (Fig. 5a). Confocal microscopy analysis revealed a clear separation between the three target tissues, enabling simultaneous imaging of the neuronal, neuroblast, and surface glia compartments in the same third-instar larval CNS (Fig. 5b, c). Notably, in all cases the level of expression appeared to be qualitatively similar to single TALE–VAS experiments, indicating that parallel activation of multiple transgenes has no influence on the performance of each individual TALE–VAS pair (compare Figs. 2 and 5 panels). The ability to drive specific and robust parallel transgene expression is instrumental for understanding complex interactions within the nervous system, such as the interplay between different classes of neuron, glial cells and neurons, or muscles and neurons. To illustrate the potential of employing TALE–VAS binary expression systems for such studies, we sought to image at high-magnification a single segmental nerve and the larval neuromuscular junction (NMJ). To this end, we crossed the triple VAS responder line either with the $elav^{1.8kb}$-$TALE_1$, $ase^{0.8kb}$-$TALE_3$, $repo^{1.9kb}$-$TALE_4$ triple TALE, or with a new $elav^{1.8kb}$-$TALE_1$, $mhc^{2.4kb}$-$TALE_3$, $repo^{1.9kb}$-$TALE_4$ line, respectively. This approach enabled specific differential labelling of an axon bundle ensheathed by a glial cell in the segmental nerve, and of the neuronal pre-synaptic terminal and the post-synaptic muscle field at the NMJ (Fig. 5d, e).

Finally, to validate our off-target bioinformatics analysis in vivo, we compared the activity of $TALE_3$ and $TALE_4$ drivers at on-target VAS sites relative to putative off-target sites located upstream and in close proximity of endogenous TSS (Supplementary Fig. 13a, Supplementary Data 1). We focused on two candidate genes, *CG16890* and *CG5613*, which are expressed at low to moderate endogenous levels in third larval instars (modENCODE Tissue Expression Data; FlyBase[19]). Transcript levels of each corresponding transgene (Cerulean and mCherry) and endogenous gene (*CG16890* and *CG5613*) were measured by RT-qPCR in third larval instars of wild-type, homozygous triple TALE-driver, triple VAS-responder, and triple TALE-driver > triple VAS-responder lines (Supplementary Fig. 13b). As expected, this analysis revealed very potent and specific activation of each responder transgene by the corresponding TALE drivers.

In contrast, no apparent changes in expression were detected between control and experimental conditions at the two endogenous off-target loci (Supplementary Fig. 13b).

## Discussion

Manipulating several tissues in parallel by consecutive expression of distinct transgenes has been challenging so far. We reasoned that this limitation could be addressed by integrating programmable DNA binding proteins in the design of binary expression systems. To this end, we have developed a highly versatile and multiplexable TALE-based driver–responder technology in *Drosophila*, which could facilitate future studies entailing complex tissue-specific manipulations. Using this platform, we demonstrate that three different transgenes can be expressed in various neighbouring tissues in a highly specific and robust manner. Furthermore, we show that TALE–VAS lines can be combined with orthogonal binary expression systems (e.g., GAL4–UAS) without any apparent crosstalk between the platforms, paving the way for future cell–cell interaction studies.

A notable feature of the TALE–VAS system is its apparent generic compatibility with fly development and adult survival. In contrast to the first generation of QF lines, which were lethal when expressed in a broad pattern (e.g., pan-neuronal under an *elav* enhancer)[6], all crosses ranging from a single TALE–VAS pair to triple TALE–VAS combinations generated viable offspring. TALE–VAS flies could also be reared as stable homozygous stocks (e.g., $elav^{1.8kb}$-$TALE_1$; $VAS_1$-Citrine-HA, $VAS_4$-FLAG-Cerulean). Furthermore, pan-neuronal expression using $elav^{1.8kb}$-$TALE_1$ or pan-muscular expression with $mhc^{2.4kb}$-$TALE_3$ resulted in healthy offspring. Although we have not directly quantified lifespan, these animals did not display any obvious reduction in viability.

Since the TALE–VAS pairs reported in this study displayed low level of background activity in certain tissues, they may not be directly suitable for the expression of lethal transgenes. However, the inherent unrestricted multiplexing capacity of the TALE–VAS system provides a unique opportunity to generate background-free driver–responder pairs, thus extending the range of future implementations to any transgene of interest. Furthermore, the effector domains fused to the TALE array could also be easily replaced in subsequent iterations of the technology to enable an even broader spectrum of applications. For example, TALEs could be fused to GAL4 or QF activation domains, which can be silenced by GAL80[20] or QS[5], respectively. This would render the

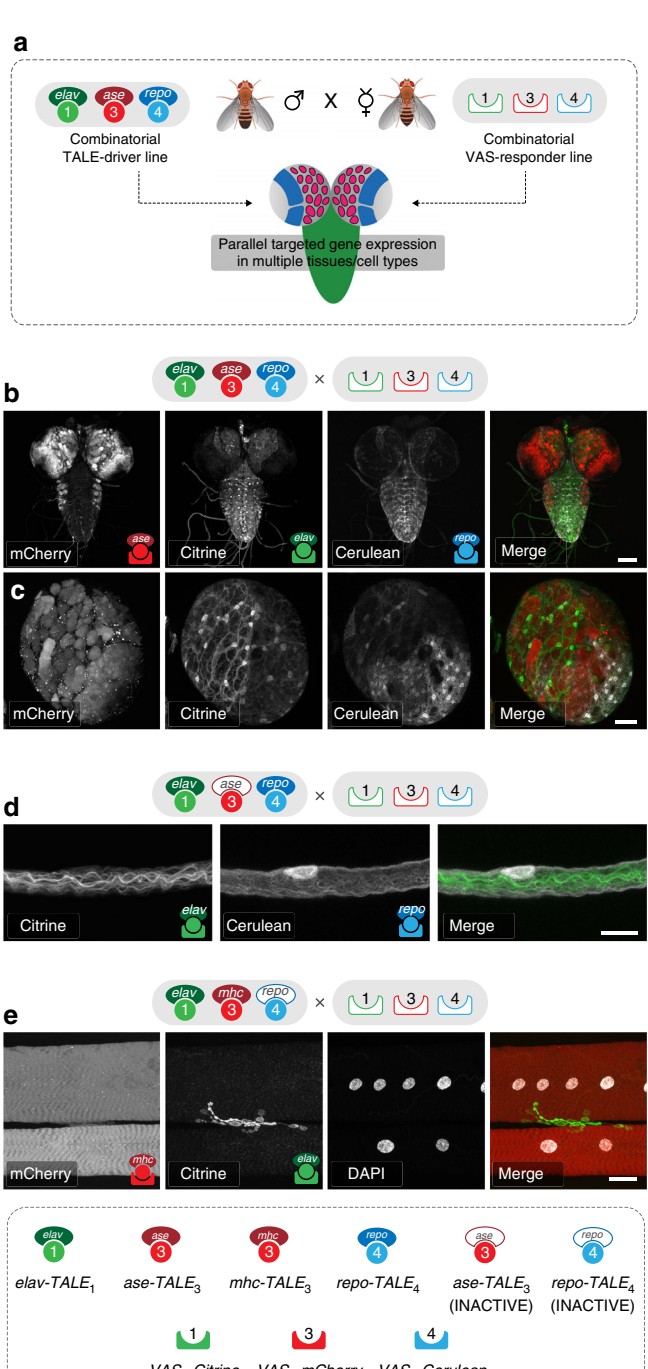

**Fig. 5** High-specificity multiplex transgene expression in *Drosophila* nervous system. **a** Diagrammatic representation of the experimental framework used for TALE–VAS-mediated parallel transgene expression in multiple tissues in vivo. Tissue-specific expression of three independent transgenes is achieved in the F0 generation by crossing combinatorial driver and responder lines. **b** Simultaneous expression of mCherry in neuroblasts (red), Citrine in neurons (green), and Cerulean in surface glia cells (white) by three independent TALE–VAS pairs in a single *Drosophila* larval CNS ($elav^{1.8kb}$-$TALE_1$, $ase^{0.8kb}$-$TALE_3$, $repo^{1.9kb}$-$TALE_4$ > $VAS_1$-Citrine-HA, $VAS_3$-V5-mCherry, $VAS_4$-FLAG-Cerulean; dorsal view of unfixed CNS). **c** High-magnification imaging of the right optic lobe from the same genotype in **b** illustrating the three distinct populations of labelled neighbouring cells (neuroblasts, neurons, and glial cells). **d** High-magnification imaging of a larval segmental nerve showing an axonal bundle (Citrine, green) ensheathed by a glial cell (Cerulean, white). The $ase^{0.8kb}$-$TALE_3$ driver is inactive in this tissue. **e** The NMJ innervating muscle 6 and 7 from $elav^{1.8kb}$-$TALE_1$, $mhc^{2.4kb}$-$TALE_3$, $repo^{1.9kb}$-$TALE_4$ > $VAS_1$-Citrine-HA, $VAS_3$-V5-mCherry, $VAS_4$-FLAG-Cerulean larvae. The post-synaptic muscle field is labelled by mCherry expression (red) and the neuronal pre-synaptic terminal by Citrine (green). The $repo^{1.9kb}$-$TALE_4$ driver is inactive in this tissue. Scale bars = 100 μm (**b**), 25 μm (**c**, **e**), 10 μm (**d**)

TALE–VAS system suitable for clonal studies employing the mosaic analysis with a repressible cell marker strategy[21].

Finally, we envision that in future implementations of this technology, repertoires of TALE drivers could be generated based on available GAL4 driver libraries with well characterised expression patterns in development or adult tissues (i.e., the nervous system)[22,23]. In principle, this could be readily achieved by directly replacing the GAL4 sequence in existing drivers with specific TALE sequences using the InSITE system, Trojan-MiMICs, or the CRISPR/Cas9 toolbox[24–26]. Considering the successful use of TALE transcription factors in vertebrate systems, it is plausible that the TALE–VAS system will also be applicable to other model organisms, where the use of GAL4–UAS has been limited so far[27,28].

## Methods

**VAS design and TALE assembly.** All VAS sequences designed were 20 nt long to ensure high specificity and robust TALE binding. Random sequences were generated using the online Fasta sequence toolbox FaBox 1.41 (http://users-birc.au.dk/biopv/php/fabox/random_sequence_generator.php)[29] taking into account the rules for optimal TALE design[30]: following an invariable T at position 0 that is not bound by the TALE array, avoid T at position 1, avoid A at position 2, and the last 3′ nucleotide should be a T. Since A at position 1 is frequent in naturally occurring TALEs, all VAS sequences started with a 5′ A and ended with a 3′ T nucleotide (Supplementary Data 2). To avoid the possibility of genomic on-target binding, each candidate sequence was then queried against the *Drosophila* genome using the National Center for Biotechnology Information's BLAST similarity search algorithm[31]. Corresponding 19 monomer long TALE arrays (Supplementary Data 2) were assembled using the Golden Gate TALEN and TAL Effector Kit 2.0 (gift from Daniel Voytas and Adam Bogdanove, Addgene kit #1000000024) as previously reported[30]. Briefly, for each position in the TALE array a module plasmid encoding the corresponding RVD was selected from the kit library (Supplementary Data 2). All RVD plasmids for positions 1–10 were assembled in the array plasmid pFUS_A, while those corresponding to positions 11–18 were assembled in pFUS_B8. The two array plasmids and the last RVD (position 19) were subsequently combined in the final backbone plasmid.

**Generation of TALE driver constructs.** Assembled TALEs were subcloned into either pJC-TALE-VP64-SV40 (gift from David Stern, Addgene plasmid #46147)[12] or pJC-TALE-VP64-P10 (Supplementary Fig. 1). pJC-TALE-VP64-P10 was generated from pJC-TALE-VP64-SV40 as follows: The TALE-VP64-SV40 sequence was removed from the pJC-TALE-VP64-SV40 plasmid by XhoI and EcoRI digest. The P10 terminator sequence was then amplified from plasmid pJFRC81 using primers *P10_InFusion_Fwd* and *P10_InFusion_Rev* (Supplementary Data 3), and the TALE-VP64 construct was amplified from plasmid pJC-TALE-VP64-SV40 using primers *TALE-VP64_InFusion_Fwd* and *TALE-VP64_In-Fusion_Rev*. Finally, the three components were re-assembled into the new pJC-TALE-VP64-P10 vector using the In-Fusion HD Cloning Kit (Clontech, 639648). To generate plasmid pJC-20×UAS-GAL4-SV40, the GAL4 sequence was amplified from plasmid pChs-GAL4 (gift from A. Bassett) with primers *IF-pJC-GAL4-Fwd* and *IF-pJC-GAL4-Rev*. The amplicon was then digested with XhoI and XbaI and inserted into pJC-TALE-VP64-SV40 replacing the TALE-VP64 sequence. pJC-20×UAS-GFP-SV40 was obtained through excision of the EGFP sequence from plasmid pJFRC81 with XhoI and XbaI, and insertion of the resulting fragment into pJC-TALE-VP64-SV40 replacing the TALE-VP64 sequence. pJC-10×UAS-GFP-SV40 was generated by removing 10×UAS from the plasmid by restriction enzyme cloning (HindIII, NheI) and inserting annealed oligonucleotides *HindIII-20bp-NheI-Fwd* and *HindIII-20bp-NheI-Rev*.

**Drosophila enhancers.** To direct the activity of the TALE–VAS system to specific tissues in flies, four different enhancer elements were cloned upstream of the hsp70 minimal promoter in the TALE₁, TALE₃, and TALE₄ vectors (Supplementary Fig. 8): (1) The ~2.4 kb myosin heavy chain (*mhc*) enhancer element[32], located between position −283 bp and +2115 bp relative to the TSS, referred to in this paper as *mhc^2.4kb* enhancer; (2) The ~1.8 kb embryonic lethal abnormal vision (*elav*) enhancer element[33], representing the region between position +247 bp and −1604 bp including the 333 bp region defined as the minimal promotor for neural *elav* expression, referred to in this paper as *elav^1.8kb* enhancer; (3) The ~0.8 kb asense (*ase*) enhancer element (ase10 enhancer at http://cispatterns.ninds.nih.gov/app/), spanning the region from position −38 to −839 bp, referred to in this paper as *ase^0.8kb* enhancer; (4) The first 1.9 kb of the −4.2 kb reversed polarity (*repo*) enhancer element (−1.9 kb enhancer in ref. [34]), referred to as *repo^1.9kb* enhancer in this paper.

The *mhc^2.4kb* enhancer was amplified from plasmid pFS209_CaSpeR_Mhc-GAL4 (a gift from F. Schnorrer[32]) using primers *mhc-BsmBI-HindIII-Fwd* and *mhc-FseI-Rev*. The *ase^0.8kb* (ase10), *elav^1.8kb*, and *repo^1.9kb* enhancer elements were

amplified from wild-type fly genomic DNA (*w^1118*) with primers *ase10-BsmBI-HindIII-Fwd*, *ase10-FseI-Rev*, *elav-BsmBI-NheI-Fwd*, *elav1.8kb-BsmBI-HindIII-Rev*, *repo-BsmBI-HindIII-Fwd*, and *repo-BsmBI-NheI-Rev*, respectively. The hsp70 promoter was amplified either from pJC-20×UAS-TALE₁-VP64-P10 using primers *Hsp70-BsmBI-NheI-Fwd* and *Hsp70-BsmBI-BglII-Rev* or from pJC-20×UAS-TALE₃-VP64-P10 using primers *Hsp70-FseI-Fwd* and *Hsp70-BsmBI-BglII-Rev*. The different enhancer elements were joined with the hsp70 promoter amplicon (generating either a unique FseI or NheI site between enhancer and promoter) by fusion PCR followed by BsmBI and BglII digest. The three different TALE plasmids (pJC-20×UAS-TALE₁,₃,₄-VP64-P10) were digested with HindIII and BglII (which removes the 20×UAS and the existing hsp70 promoter) and the respective fusion PCR fragments were inserted by standard ligation.

**VAS responders.** VAS responder constructs were based on the pJFRC81 vector (gift from Gerald Rubin, Addgene plasmid #36432)[13]. To generate 1×, 3×, 5×, and 10×VAS constructs, the sequence encoding 10×UAS was removed from pJFRC81 with HindIII and NheI, and replaced with the respective numbers of VAS copies obtained either through annealed DNA oligos or amplification of previously cloned sequence repeats. The Citrine, Cerulean, and mCherry genes were amplified from plasmids (gift from T. Sauka-Spengler) using primers *VAS-1_Citrine_Fwd*, *VAS-1_Citrine_Rev* (containing the HA-tag), *VAS-2_FLAG-Cerulean_Fwd*, *VAS-2_Cerulean_Rev*, *VAS-3_mCherry_Fwd*, and *VAS-3_mCherry_Rev*. The Citrine and mCherry amplicons were further extended by fusion PCR. Both were fused to the Syn21 sequence using the *VAS-1_Syn21* oligo or the *VAS-3_Syn21* oligo (containing the V5-tag) in conjunction with the primers *VAS-1_Syn21_Fwd* and *VAS-1_Citrine_Rev*, or *VAS-3_Syn21_Fwd* and *VAS-3_mCherry_Rev* to generate fragments Syn21-Citrine-HA and Syn21-V5-mCherry, respectively. For the generation of the Citrine, Cerulean, and mCherry responders, the P10 terminator was amplified from plasmid pJFRC81_3×VAS₁-GFP with primers *VAS-1_HA-P10_Fwd* and *VAS-1_P10_Rev*, *VAS-2&3_P10_Fwd* and *VAS-2_P10_Rev*, or *VAS-2&3_P10_Fwd* and *VAS-3_P10_Rev* resulting in fragments VAS-1-HA-P10, VAS-2-P10, and VAS-3-P10, respectively. For the Cerulean responder, Hsp70 was amplified from plasmid pJFRC81_3×VAS₂-GFP with primers *VAS-2_hsp70_Fwd* and *VAS-2_hsp70_Rev*. This fragment was further extended by fusion PCR using the *VAS-2_Syn21* oligo and primers *VAS-2_hsp70_Fwd* and *VAS-2_FLAG_Rev* to obtain fragment Hsp70-Syn21. The *mhc* splice donor (Mhc-SD) and acceptor (Mhc-SA) sequences were amplified from wild-type genomic DNA (Oregon R) using primers *MHC_SD_Fwd*, *MHC_SD_Rev* and *MHC_SA_Fwd*, *MHC_SA_Rev*, respectively. The Citrine responder (pJFRC81_3×VAS₁-Syn21-Citrine-HA-P10) was assembled from plasmid pJFRC81_3×VAS-1-GFP-P10 digested with BglII and EcoRI, and fragments Syn21-Citrine-HA, VAS-1-HA-P10, Mhc-SD by in-fusion cloning (Clontech, 639648). The Cerulean responder was assembled from plasmid 3×VAS-2-GFP-P10 digested with AatII and EcoRI, and fragments Hsp70-Syn21, FLAG-Cerulean and VAS-2-P10 by in-fusion cloning. Subsequently, the 3×VAS₂ sequence in pJFRC81_3×VAS₂-Syn21-FLAG-Cerulean-P10 was exchanged for the 3×VAS₄ sequence from pJFRC81_3×VAS₄-GFP-P10 by restriction enzyme cloning resulting in pJFRC81_3×VAS₄-Syn21-FLAG-Cerulean-P10. The mCherry responder was assembled from plasmid pJFRC81_3×VAS-3-GFP-P10 (digested with BglII and EcoRI) and fragments Syn21-V5-mCherry, VAS-3-P10, Mhc-SA by in-fusion cloning (Clontech, 639648) resulting in pJFRC81_3×VAS₃-Syn21-V5-mCherry-P10.

For the generation of a Citrine-Cerulean double responder vector, the 3×VAS₄-hsp70-Syn21-FLAG-Cerulean-P10 cassette was amplified from plasmid pJFRC81_3×VAS₄-Syn21-FLAG-Cerulaen-P10 using primers *3×VAS₄-Cerulean-Fwd* and *3×VAS₄-Cerulean-Rev*. The resulting cassette was then inserted into plasmid pJFRC81_3×VAS₁-Syn21-Citrine-HA-P10 digested with EcoRI by in-fusion cloning (Clontech, 639648). A list of all primers and DNA oligonucleotides used in this study is provided in Supplementary Data 3.

**Drosophila S2 cell culture experiments.** The *Drosophila* S2 cell line containing at least one stably integrated pMT-GAL4 construct was a gift from A. Bassett. Cells were cultured in *Drosophila* Schneider's media (Gibco, 21720-024) supplemented with 10% (v/v) heat inactivated fetal bovine serum (Gibco, 10500-056) and 0.5% (v/v) penicillin streptomycin (Gibco, 15140-122) at 26 °C and passaged every 3–4 days.

For all transfection experiments, cells were seeded at a density of $1 \times 10^6$ cells per ml in 6, 12, or 24 well cell culture plates in full media 24 h prior to transfection or on the day of transfection. Two different transfection reagents were used according to the manufacturers' protocols: Effectene (Qiagen, 301425) at a ratio of 1:10 DNA to Effectene (1 µg DNA:10 µl Effectene, 10 min incubation) in full media; FuGENE (Promega, E2311) at a ratio of 1:3 DNA to FuGENE (1 µg DNA:3 µl FuGENE, 40 min incubation) in serum-free media. GAL4 expression was induced 24 h after transfection by addition of a 100× CuSO₄ stock solution (100 mM) to a final concentration of 0.1–0.7 mM CuSO₄ per well. EGFP expression was assessed 12–48 h after induction either by flow cytometry or confocal microscopy.

For flow cytometry measurements, S2 pMT-GAL4 cells were harvested by gently pipetting a stream of media over the well surface, and the resulting suspension was transferred to a 1.5 ml tube and centrifuged at $1000 \times g$ for 3 min. Media was removed and the cells washed in 1× PBS (phosphate buffered saline) and kept on ice. EGFP levels were measured either with a CyAn ADP flow

cytometer (DakoCytomation) or a BD LSR Fortessa cell analyser (BD Biosciences). Data analysis was performed with the Kaluza Analysis 1.3 software (Beckman Coulter), using the gating strategy described in Supplementary Fig. 14.

For confocal imaging, cells were either seeded for transfection in a 12-well culture plate on sterile, round cover slips or transferred after transfection and allowed to settle on cover slips. Cells were then fixed with 4% methanol free PFA (Electron Microscopy Sciences, 15710) for 10 min at room temperature (RT) and washed twice with 1× PBS for 5 min each. Following a single wash with 1× PBS-Triton (0.1%) for 10 min, cells were blocked in 5% normal goat serum (NGS) in 1× PBS-Triton (0.1%) for 1 h at RT. Primary antibodies were applied either at RT for 2–4 h or at 4 °C overnight. After three washes with 1× PBS-Triton (0.1%) for 10 min each, the secondary antibodies together with DAPI were applied for 2–4 h at RT or 4 °C overnight. Cells were then washed three times with 1× PBS-Triton (0.1%) for 10 min each and subsequently mounted on a microscope slide by gently lowering the cover slip with the cells facing down onto a drop of SlowFade Diamond Antifade Mount (Life Technologies, S36972). The mouse anti-GFP primary antibody (DSHB, 12A6) was used at 1:200 dilution, and the goat anti-mouse-A488 secondary antibody (Invitrogen, A11001) was used at 1:1000 dilution. Nuclear staining was performed with DAPI (Invitrogen, D1306) at 1:2000 dilution.

**Generation of transgenic animals.** Transgenic fly lines were established by phiC31 integrase-mediated germline transformation (GenetiVision, USA). Five different landing sites were chosen to allow for the generation of triple drivers and triple responders by classic recombination: attP1 and attP2[35], VK00020, VK00031, and VK00037[36] (Supplementary Fig. 9). The genomic locations of attP1 (P{CaryP} attP1) and attP2 (P{CaryP}attP2) were initially identified as chromosome 2R (cytological band 55C-D, between GM04742 and jockey) and 3L (cytological band 68A1-B2, between genes CG6310 and Mocs1), respectively[35]. These positions were later refined as follows: attP1 is located in plus orientation in band 56C1, 62 bp upstream of the gene sbb [2R:18,357,321..18,357,321, 55C4]; attP2 is in plus orientation in band 68A4, 44 bp upstream of the gene Mocs1 [3L:11,070,538..11,070,538][37]. The three VK landing sites are in plus orientation at the following genomic locations: VK00020 (PBac{y[+]-attP-9A}VK00020) on chromosome 3R (band 99F8 [3R:30,553,313..30,553,313] in the intron of the gene tmod), VK00031 (PBac{y[+]-attP-3B}VK00031) on chromosome 3L (band 62E1 [3L:2,395,487..2,395,487] in the intron of the gene CG45186), and VK00037 (PBac {y[+]-attP-3B}VK00037) on chromosome 2L (band 22A3 [2L:1,582,820..1,582,820] in the intron of the gene haf) (Supplementary Fig. 9)[36]. All chromosomal locations in square brackets are based on FlyBase (FB2017_02, released 18 April 2017). For each TALE and VAS construct two to three fly lines were obtained each representing an independent integration event.

In addition to the TALE–VAS lines, the following fly stocks were used in this study: UAS-mCD8-mCherry (BL27391), UAS-mCD8-GFP/CyO. Orgeon R or w[1118] were used as wild-type. ase-GAL4 was a gift from Ji-Long Liu. However, the exact origin of this line could not be retraced. The most commonly used ase-GAL4 lines[38,39] are based on the ~2 kb upstream enhancer element described in Jarman et al.[40] or were generated by cloning ~2–4 kb long genomic fragments flanking the ase gene (Janelia lines)[41]. All these lines contain substantially larger genomic fragments compared to the 0.8 kb sequence used for the ase-TALE_3. Flies were reared on standard food at 18 or 25 °C. All crosses were performed at 25 °C unless indicated otherwise. Compound stocks, triple driver, and triple responder lines were generated by classic recombination and standard genetic techniques (Supplementary Data 4).

**Larval dissections and immunohistochemistry.** Wandering third instar larvae were dissected one at a time in a 9-well clear glass spot plate in 1× PBS. The anterior end (approximately one-third) of the larva was cut off with scissors and inverted using a fine forceps. After removing any obstructing fat body and the salivary glands, the exposed CNS was freed from the carcass. The imaginal discs were subsequently removed with a fine needle.

For immunohistochemistry, larvae were dissected as described above, except the brains were left attached to the carcasses. All protocol steps were carried out at RT unless noted otherwise with gentle agitation, and washes were performed in PBS-Triton (0.1%) for 20 min each. Specimens were fixed in 4% methanol-free PFA (Electron Microscopy Sciences, 15710) in PBS-Triton (0.1%) for 30 min, washed three times and then blocked in PBS-Triton (0.1%) with 5% NGS for 1 h. Primary antibody staining was carried out in PBS overnight at 4 °C. Samples were then washed three times, incubated with the secondary antibodies, phalloidin-A488 and the nuclear stain in 1× PBS for 3–4 h, and washed again three times. Brains were subsequently separated from the carcasses and the imaginal discs were removed. For the analysis of eye discs, the discs were carefully severed from the brain at this step. The brains or eye discs were then mounted on a microscope slide in SlowFade Diamond Antifade Mountant (Life Technologies, S36972).

The following primary and secondary antibodies were used throughout the study: rabbit anti-HA at 1:100 (BETHYL Laboratories, A190-108A), rat anti-Elav at 1:100 (DSHB, 7E8A10), mouse anti-Pros at 1:100 (DSHB, MR1A), mouse anti-Repo at 1:100 (DSHB, 8D12), rabbit anti-GFP at 1:200 (Invitrogen, A11122); goat anti-rat-A568 at 1:1000 (Invitrogen, A11077), goat anti-rabbit-A647 at 1:1000 (Invitrogen, A21244), goat anti-mouse-A488 at 1:1000 (Invitrogen, A11001), and goat anti-mouse-A568 at 1:1000 (Invitrogen, A11004). Nuclear staining was

performed with DAPI at 1:2000 (Invitrogen, D1306). Filamentous actin was labelled with phalloidin-A488 at 1:1000 (Invitrogen, A12379).

**Light microscopy and confocal imaging.** For imaging immobilised wandering third larval instars, animals were collected from the vials, rinsed with water to remove food remnants, and placed dorsal side up on a microscope slide between bridges made of cover slips (22 × 22 mm). A large cover slip (22 × 50 mm) was placed on top and the space in between filled with water. The strong adhesive force of the water holds the large cover slip in place, squeezing the larva, and thereby rendering it immobile. Images were acquired on a MVX10 stereo microscope (Olympus) equipped with a MV PLAPO 1× objective (Olympus) and an AxioCam HRc (Zeiss) operated by the AxioVision 4.8 software (Zeiss).

For confocal imaging of freshly dissected brains, third larval instar brains were transferred with a fine forceps to a drop of PBS on a microscope slide between two bridges made of coverslips (22 × 22 mm). A large coverslip (22 × 50 mm) was placed on top of the bridges and PBS added until the space between the bridges containing the brain was completely filled. Confocal images were acquired on a Zeiss 780 inverted microscope equipped with the Zen software. Low magnification images of entire larval brain were obtained with an EC Plan-Neofluar 10×/0.30 M27 objective, while high magnification images of S2 cells, selected brain areas, NMJ, and nerves were acquired either with a LD C-Apochromat 40×/1.1 W Korr M27 or Plan-Apochromat 63×/1.40 Oil DIC M27objective.

**Confocal image processing.** 3D confocal stacks were converted into 2D images using the maximum intensity projection tool provided by the Zen software (Zeiss). Images were exported from the Zen or AxioVision 4.8 software (Zeiss) as TIFs in RGB colour mode and processed in Adobe Photoshop CS5 (Adobe). For publication purposes, brightness and contrast was adjusted, where necessary, using the levels function. In case images were acquired using the same microscope settings, processing was performed using exactly the same parameters. Cropping was carried out using the rectangular marquee tool without modifying the image resolution. The cyan channel was pseudo-coloured by adding the information from the blue channel to the green channel. All figures were assembled in Adobe Illustrator CS6 (Adobe). Bar graphs and statistical analyses were performed with GraphPad Prism 7 (GraphPad).

**Temperature-dependent TALE–VAS experiments.** Multiple crosses between 10 males of the elav[1.8kb]-TALE_1 driver and 10 virgin females of the triple responder (VAS_3-V5-mCherry, VAS_1-Citrine-HA, VAS_4-FLAG-Cerulean) were set up at 18 and 25 °C. Wandering third larval instars were picked from vials and dissected imaginal discs and brains were analysed either by confocal microscopy (immunohistochemistry) or RT-qPCR. Maximum intensity projections of brains or eye discs were generated with the Zen software (Zeiss) and exported as TIFs. For the analysis of Citrine-HA transgene expression in eye discs, an area within the retina was cropped using Adobe Photoshop CS5 (Adobe). The mean pixel intensity of each channel within this area was then calculated using the measure tool in the ImageJ package (http://fiji.sc/Fiji). To account for differences in the staining efficiency, signal intensity (anti-HA staining) was normalised to Elav levels.

**TALE off-target analysis.** To identify possible genomic off-targets, a window spanning 2 kb upstream and 2 kb downstream from each Drosophila TSS annotated in the UCSC transcriptome database was analysed. The R package "TxDb.Dmelanogaster.UCSC.dm6.ensGene" was employed to map TSS using the "BSgenome. Dmelanogaster.UCSC.dm6" Drosophila genome release. Possible off-target sites were identified by scanning both strands within each 4 kb window for sequences similar to VAS_{1-4} using the R function "vmatchPattern" allowing up to 4 nt mismatches.

To assess the effect of potential off-target binding on endogenous gene regulation, homozygous stocks were reared at 25 °C, third larval instars were picked from the wall of the culture vessels, and processed for RT-qPCR. The wild-type (w[1118]) and the triple VAS-responder line (VAS_1-Citrine-HA, VAS_3-V5-mCherry, VAS_4-FLAG-Cerulean) served as non-TALE expressing controls, while the triple TALE driver line (elav[1.8kb]-TALE_1, ase[0.8kb]-TALE_3, repo[1.9kb]-TALE_4) and the triple TALE-driver > triple VAS-responder line were used to analyse on-target and off-target activation.

**RT-qPCR analysis.** For the analysis of temperature impact on TALE activity, third larval instar brains were dissected one at a time as described above in DEPC (Sigma, D5758) treated 1× PBS and immediately transferred to ice cold RLT lysis buffer (RNeasy Mini Kit, Qiagen, 74106) containing 1% (v/v) β-Mercaptoethanol (Sigma, M6250). Samples were then vortexed for 20 s at 2000 rpm to facilitate the lysis reaction, and stored in lysis buffer at −80 °C until RNA extraction. Three biological replicates each containing ten brains were processed for every temperature data point. For the off-target analysis, total third larval instars were homogenised in ice cold RLT lysis buffer containing 1% (v/v) β-Mercaptoethanol with a motor pestle. After vortexing for 20 s at 2000 rpm, samples were immediately used for RNA extraction. Three biological replicates each containing five third larval instars were prepared for each genotype. Total RNA was extracted with the RNeasy Mini Kit (Qiagen, 74106). For each sample, 600 ng (temperature

experiment) or 1 µg (off-target analysis) of total RNA was reverse transcribed into cDNA using the QuantiTect Reverse Transcription Kit (Qiagen, 205313). Quantitative real-time PCR (qPCR) was performed on a CFX384 real-time system (Bio-Rad) using the SsoAdvanced Universal SYBR Green Supermix Kit (Bio-Rad, 1725272). Each biological replicate was run in technical triplicates. TALE mRNA levels were measured using qPCR primers qPCR-TALN-Fwd and qPCR-TALN-Rev both binding within the N-terminal part of the TALE sequence. To avoid unspecific amplification of similar fluorophores, Citrine mRNA levels were quantified using a forward qPCR primer binding to the Citrine sequence (qPCR-Citrine-Fwd) and a reverse qPCR primer binding to the HA-tag (qPCR-HA-Rev). Similarly, primers qPCR-FLAG-Fwd and qPCR-Cerulean-Rev, as well as qPCR-V5-Fwd and qPCR-mCherry-Rev were used to measure Cerulean and mCherry mRNA levels (Supplementary Data 3). Finally, primers qPCR-CG16890-Fwd and qPCR-CG16890-Rev or qPCR-CG5613-Fwd and qPCR-CG5613-Rev were used for the analysis of candidate off-target genes CG16890 or CG5613, respectively. Expression levels were first normalised to Actin5C mRNA levels (primers: qPCR-Act5C-Fwd and qPCR-Act5C-Rev) using the $\Delta$Ct method (e.g., transcript level of TALE = $2^{(Ct_{Actin5C}-Ct_{TALE})}$) and then normalised to the TALE expression.

**Data availability**. Relevant TALE-VAS *Drosophila* stocks described in this study are available from the Bloomington Drosophila Stock Center (http://flystocks.bio.indiana.edu/). Plasmids and constructs described in this study are available from Addgene (http://www.addgene.org/Tudor_Fulga/).

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

## Acknowledgements

We are grateful to Andrew Bassett for providing the S2 pMT-GAL4 cell line, Tatjana Sauka-Spengler for the Citrine, mCherry and Cerulean constructs, Frank Schnorrer for the mhc enhancer plasmid, and Ilan Davis, Ji-Long Liu, and the Bloomington Drosophila Stock Center (NIH P40OD018537) for providing fly stocks. We would like to thank Kevin Clark and Paul Sopp (WIMM FACS Core Facility) for assistance with flow cytometry measurements and technical expertise, and Christoffer Lagerholm (Wolfson Imaging Centre Oxford) for help with confocal imaging. We are grateful to Ilan Davis, Aron Szabo, Quentin Ferry, Yale Michaels, and Toni Baeumler for providing critical comments on the manuscript. M.T. was supported by MRC (#G0902418) to T.A.F. and BBSRC (#BB/L010275/1). G.A. was supported by a fellowship from the Malaysian Ministry of Education. E.L. was supported by a fellowship from Princeton University and MRC (#G0902418) to T.A.F. D.J.H.F.K. is supported by a CIHR Postdoctoral Fellowship (#201511MFE-358733-204362) and BBSRC (#BB/N006550/1). T.A.F. is supported by MRC (#G0902418) and BBSRC (#BB/L010275/1).

## Author contributions

M.T., G.A. and T.A.F. conceived the study and designed the experiments. G.A. cloned the TALE and VAS vectors and carried out most of the S2 cell experiments. M.T. performed the TALE-GAL4 comparison in S2 cells and cloned the fly enhancers with help from E.L. M.T. performed all in vivo experiments. Y.T. and M.F. generated the TALE-VAS

*Drosophila* lines. D.J.H.F.K. carried out the bioinformatics analysis. M.T., G.A. and T.A.F. analysed the results. M.T. and T.A.F. generated the figures and wrote the manuscript.

## Additional information

**Competing interests:** Y.T. and M.F. are employees of GenetiVision Corporation, a company that provides *Drosophila* transgenic services. The remaining authors declare no competing financial interests.

