## [Peer Review File · Nature Communications]

Reviewers' comments:

Reviewer #1 (Remarks to the Author):

In this manuscript, Toegel et al. describe a new binary expression system specifically designed for the purpose of gaining genetic access to multiple cell types simultaneously. Currently, such binary systems (Gal4-UAS, LexA, QF) are limited by the fact that each activator binds to a single operator sequence – thus, researchers who wish to genetically access two cell types must combine two orthogonal systems. To get around this issue, the authors generate customized TALE proteins that target four different sequence-specific transcriptional activators that target four unique binding sequences (“VAS” sequences), of which three are highly effective. And presumably, an essentially limitless number of unique TALE-VAS pairs can be designed in the future. In theory, the system is limited primarily by the technical difficulty of bringing together all of the transgenes into a single fly.

This manuscript describes the development and characterization of the TALE-VAS system in cell culture and in vivo in *Drosophila*. The TALEs activate VAS-reporter genes at levels similar to the Gal4-UAS system, and each TALE-VAS pair appears to have no discernable cross-talk with each other, or with the Gal4-UAS system. Importantly, expression of multiple TALE-VAS pairs in vivo does not appear to cause toxicity, and in the final figure, the authors demonstrate successful genetic labeling of three distinct cell types in a single fly. In addition, the authors present several caveats of the system, including an unexpected temperature-dependence, and the fact that their reporter lines have some background expression patterns.

In my opinion, this research represents a potentially very useful new tool that will be of interest to many researchers in the *Drosophila* field, and likely in other model organisms as well. The paper is well-written, clear, and straight-forward, and I recommend it for publication in *Nature Communications* with minor revisions. Specifically, the authors should more explicitly address a few potential caveats with the system that are presented in their data, but are not yet directly addressed in the main text.

Comments:

1) More explicit characterization of VAS leakiness in the absence of a TALE. The authors provide several demonstrations of TALE > VAS:EGFP activation over control levels (a control TALE), including Figure 1c and e, and Supplemental Figures 3-4. However, the flow cytometry data shown in Supplemental Figures 3-4 shows a fair amount of activation in the control conditions (although clearly far below the levels seen in the presence of the correct TALE). I recommend that the authors include an additional control experiment showing EGFP levels in the absence of a VAS:EGFP reporter construct, to give a direct read-out of how much background EGFP levels are driven by leakiness from the VAS promoter. While such leakiness issue may not affect one's ability to label cells with fluorophores (as demonstrated nicely in this paper), it could certainly have effects when using cellular manipulations that are potent at low levels of expression, and would be best to characterize.

2) Comment on the cause of imperfect overlap of between VAS signal and endogenous gene expression (Figure 2). The authors should clarify more explicitly in the text on their explanation for why there is <100% overlap of the TALE>VAS expression and endogenous gene expression – particularly notable for Repo. The explanation is given in the Methods section (that the enhancer fragments are not expected to fully recapitulate endogenous expression, which is perfectly reasonable), but I recommend moving a note on this into the main text to clarify that it isn't necessarily an inherent bug of TALE-VAS approach.

3) Direct comparison of aseTALE-VAS and aseGal4-UAS? In Figure 3a, the authors perform a seemingly direct comparison between an ase-Gal4 line and aseTALE-VAS – however, the

magnification provided does not allow the reader to assess how much overlap there is between the two systems at the level of individual cells. I recommend a higher magnification panel, and ideally a quantification of how much overlap there is between the two systems. In addition, I can't figure out from the methods whether or not the aseGal4 and ase-TALE actually represent identical enhancer fragments (unless I missed it, I believe the aseGal4 isn't sourced in the methods section).

4) The authors should more clearly explain why they chose to use S2 cells that stably express Gal4. They imply that this was simply a background feature of the cells they use, but they should more clearly explain why it's there in the main text.

Reviewer #2 (Remarks to the Author):

The authors present an intriguing strategy for multiplexed binary expression in *Drosophila* using rationally designed TALE-AD fusions to drive expression of multiple reporter genes independently in the same transgenic animal. They then present and characterize a somewhat limited implementation of the strategy. Expression levels and independence of three TALE-VP64 artificial TFs (and their cognate targets, here termed "VAS elements" for "variable activating sequences") are first tested in transiently transfected S2 cells, and then further characterized in flies.

The demonstration of orthogonality of the three tested TALE-VAS combinations, among themselves and with the GAL4-UAS system, is careful and convincing. Additional evaluation of the temperature dependence of TALE-VP64 activation is a very nice addition, and contributes greatly to the usefulness of the system.

My principal complaints about the MS are with the rather cavalier treatment of off-target and ectopic effects. It would perhaps be beyond the scope of the study to suggest ChIP-seq for the artificial TFs, or RNA-seq of TALE-VP64-expressing animals, but the considerable literature on off-target effects (see e.g. doi: 10.1111/febs.13760 for a recent review in the context of engineered nucleases) should certainly be addressed. It should be noted in this context that the authors do observe viability and lack of a visible phenotype from expression of their artificial TFs, which helps to allay these concerns somewhat, although the subtle effects that this analysis would miss could complicate the interpretation of the more advanced experiments discussed below. Some followup experiments would be straightforward to perform using no techniques not already in place: the authors suggest, for example, that some observed ectopic activation could be due to the use of a single landing site for TALE-VP64 fusions, but do not test this by comparison with other landing sites.

These concerns about specificity, ectopic activation, and off-target effects are amplified by the consideration that the authors use their system only to drive fluorescent reporter proteins, while a principle application of binary expression systems to date has been to drive the expression of genes that would otherwise be lethal and prevent the establishment and maintenance of useful stocks. The authors indirectly allude to this very use case with the beginning of their Discussion: "Manipulating several tissues in parallel by consecutive expression of distinct transgenes has been challenging so far." Such manipulation, as opposed to observation and characterization of tissues using fluorescent reporters, will require expression of effector transgenes, and the ectopic expression shown in Supplementary Figure 8 in the absence of TALE-VP64 expression may not permit the maintenance of stocks necessary for such experiments, while the ectopic ectodermal activation observed in multiple TALE-VAS combinations would compromise the interpretability of results from such experiments. Again, these are relatively straightforward questions to test using techniques that the authors have already demonstrated, although working around them may require the design and testing of a large number of additional TALE repeat arrays, probably

exceeding the scope of this study.

In summary, this paper is built around an excellent idea, and one that will eventually provide a useful and important toolkit to at least the *Drosophila* field, and quite likely to other model organism communities as well. To realize this toolkit would require a large amount of additional work, but they have effectively done the necessary work to establish its desirability. Discussion of a broader spectrum of applications should be couched more in the context of future directions if the authors are unable to demonstrate that such lines can currently be created.

REVIEWERS' COMMENTS:

Reviewer #1 (Remarks to the Author):

In my opinion, this research represents a potentially very useful new tool that will be of interest to many researchers in the *Drosophila* field, and likely in other model organisms as well. The paper is well-written, clear, and straight-forward, and I recommend it for publication in *Nature Communications* with minor revisions.

We thank the reviewer for their endorsement of our technology and manuscript.

Comments:

1) More explicit characterization of VAS leakiness in the absence of a TALE.....I recommend that the authors include an additional control experiment showing EGFP levels in the absence of a VAS:EGFP reporter construct, to give a direct read-out of how much background EGFP levels are driven by leakiness from the VAS promoter.

We agree with this point raised by the reviewer and have now included in the manuscript additional controls comparing background EGFP levels between untransfected cells (mock control) and cells transfected with only VAS₄-EGFP or UAS-EGFP responders alone in the absence of any driver and CuSO₄ induction (See Supplementary Fig. 7 and Figure R1). As reflected by this data, in contrast to the strong activation observed in the presence of corresponding TALE₄ or GAL4 drivers, both responders alone display low background signal relative to the mock control.

Figure R1: Characterization of VAS and UAS responders in *Drosophila* S2 cells. (a) Schematic representation of the driver-responder combinations used in this experiment. TALE driver constructs are depicted as circles (number 4 indicates TALE₄, CTR stands for TALE_{CTR}) and the GAL4 driver as a triangle. Corresponding responders have matching colours, numbers, and indentations. **(b)** Flow cytometry analysis of *Drosophila* S2 pMT-GAL4 cells transfected with various driver and responder combinations. Bar graph shows quantification of EGFP reporter expression in the presence of matching drivers (TALE₄ or GAL4), control driver (TALE_{CTR}), in the absence of any driver (responders alone), and mock transfected cells. In all cases, mean EGFP fluorescence was calculated from three biological replicates (n = 3 from one experiment, mean +/- s.d.; a.u., arbitrary units).

2) Comment on the cause of imperfect overlap between VAS signal and endogenous gene expression (Figure 2). The authors should clarify more explicitly in the text on their explanation for why there is <100% overlap of the TALE>VAS expression and endogenous gene expression – particularly notable for Repo. The explanation is given in the Methods section (that the enhancer fragments are not expected to fully recapitulate endogenous expression, which is perfectly reasonable), but I recommend moving a note on this into the main text to clarify that it isn't necessarily an inherent bug of TALE-VAS approach.

We thank the reviewer for this very helpful suggestion and have now further clarified this point and moved the following paragraph from the methods section into the results section: *“It should be noted that although these enhancers were selected to best reflect the endogenous expression patterns of their associated genes it is unlikely that the chosen sequence boundaries captured the entire set of native regulatory elements. Consequently, only a partial overlap should be expected between the TALE-drivers expression domains and those of the corresponding endogenous genes (antibody staining).”*

3) Direct comparison of aseTALE-VAS and aseGal4-UAS? In Figure 3a, the authors perform a seemingly direct comparison between an aseGal4 line and aseTALE-VAS – however, the magnification provided does not allow the reader to assess how much overlap there is between the two systems at the level of individual cells. I recommend a higher magnification panel, and ideally a quantification of how much overlap there is between the two systems. In addition, I can’t figure out from the methods whether or not the aseGal4 and ase-TALE actually represent identical enhancer fragments (unless I missed it, I believe the aseGal4 isn’t sourced in the methods section).

We appreciate the reviewer’s comment and have now added a high magnification confocal image to allow direct comparison of the overlap between the two systems (see Supplementary Fig.12 and Figure R2). This analysis clearly shows a high degree of co-localization between the ase^{0.8kb}-TALE₃ > VAS₃-V5-mCherry and ase-GAL4 > UAS-mCD8-GFP transgenes. It should be noted however, that the two transgenes localise to different cellular compartments: mCherry is cytoplasmic while mCD8-GFP is membrane bound.

Figure R2: Direct comparison of the TALE-VAS and GAL4-UAS systems in flies. High magnification confocal image of the ventral side of an optic lobe (dashed outline) from a third larval instar brain (ase^{0.8kb}-TALE₃; VAS₃-V5-mCherry; UAS-mCD8-GFP; ase-GAL4). Although both systems show varying signal intensities between individual neuroblast clusters, their overall expression pattern is comparable and overlaps in the majority of the cells.

The reviewer also correctly points out that the two enhancer fragments may not be identical. We have now clarified this aspect both in the main text: *“Although the two drivers harbour different ase enhancer elements (see Methods) and the two transgenes are localised to distinct cellular compartments (cytoplasm and membrane), their pattern of expression in neuroblast cells was predominantly overlapping (Fig. 3a and Supplementary Fig. 12)”*, and the Methods section: *“ase-GAL4 was a gift from Ji-Long Liu. However, the exact origin of this line could not be retraced. The most commonly used ase-GAL4 lines^{1, 2} are based on the ~2kb upstream enhancer element described in Jarman et al.³ or were generated by cloning ~2-4kb long genomic fragments flanking the ase gene (Janelia lines)⁴. All these lines contain substantially larger genomic fragments compared to the 0.8kb sequence used for constructing the ase-TALE₃ driver.”*

4) The authors should more clearly explain why they chose to use S2 cells that stably express Gal4. They imply that this was simply a background feature of the cells they use, but they should more clearly explain why it's there in the main text.

We thank the reviewer for pointing out this missing explanation. The original TALE plasmids (pJC-TALE-VP64) contain 20xUAS upstream of the hsp70 promoter (see Supplementary Fig. 2), enabling robust TALE expression in the presence of GAL4. To take advantage of these optimised vectors for testing our TALE drivers, we used *Drosophila* pMT-GAL4 S2 cells that express GAL4 under a CuSO₄-inducible promoter.

We have amended the following paragraphs in the Results section to clarify the use of pMT-GAL4 S2 cells: *"In order to assess the general feasibility of our approach, we first generated four different TALE drivers based on the pJC-TALE-VP64 vector⁵, which is optimised for GAL4-mediated transgene expression in Drosophila cells (Supplementary Fig. 1). The four corresponding VAS responders (VAS₁-, VAS₂-, VAS₃-, VAS₄-EGFP) were created by cloning TALE target sequences upstream of an EGFP reporter construct in the Drosophila pJFRC81 vector scaffold⁶ (Supplementary Fig. 2). To ensure high specificity and strong affinity, TALEs were designed to recognise 20 nucleotide (nt) long VAS sequences."....."These constructs were tested in Drosophila S2 cells harbouring a stably integrated Cu²⁺-inducible metallothionein promoter GAL4 (pMT-GAL4), which was used to drive TALE expression."*

Reviewer #2 (Remarks to the Author):

In summary, this paper is built around an excellent idea, and one that will eventually provide a useful and important toolkit to at least the *Drosophila* field, and quite likely to other model organism communities as well. To realize this toolkit would require a large amount of additional work, but they have effectively done the necessary work to establish its desirability.

We thank the reviewer for their endorsement of our technology and manuscript.

My principal complaints about the MS are with the rather cavalier treatment of off-target and ectopic effects. It would perhaps be beyond the scope of the study to suggest ChIP-seq for the artificial TFs, or RNA-seq of TALE-VP64-expressing animals, but the considerable literature on off-target effects (see e.g. doi: 10.1111/febs.13760 for a recent review in the context of engineered nucleases) should certainly be addressed. It should be noted in this context that the authors do observe viability and lack of a visible phenotype from expression of their artificial TFs, which helps to allay these concerns somewhat, although the subtle effects that this analysis would miss could complicate the interpretation of the more advanced experiments discussed below.

We agree with the reviewer that binding of TALE transcription factors at potential off-target sites may under certain circumstances cause undesirable effects. As pointed out by the reviewer, a genome-wide off-target analysis of by ChIP-seq or RNA-seq would be beyond the scope of the current study. However, following the reviewer's suggestion, we have now addressed this concern by including a comprehensive bioinformatics analysis of putative off-targets for each TALE:VAS pair (see revised Supplementary Table 1, Supplementary Fig. 3 and Figure R3). Furthermore, we have directly tested candidate off-target loci *in vivo* by RT-qPCR. Together, these analyses indicate that TALE₁, TALE₃ and TALE₄ drivers, which were selected for all *in vivo* experiments, display no relevant off target sites (up to three nt mismatches) in the *Drosophila* genome.

In addition, we have now substantially revised the Results, Discussion and Methods sections to clarify issues related to off-target effects, as well as the potential impact of VAS background activation and the ectopic expression observed with certain driver:responder pairs. All these changes are highlighted in blue in the revised manuscript. We agree with the reviewer that although these effects may not represent a concern when driving fluorescence reporters, they could interfere with the expression of other effector or lethal transgenes – this point is now discussed in the revised manuscript. As suggested by the reviewer, we also elaborate on the possible causes underlying the observed effects and discuss how these problems could be solved in future iterations of the technology. We agree that further work will be required to address these issues and realise the full potential of this technology (such as building a large-scale scientific

resource). However, as the reviewer astutely points out, reaching this objective will entail the design and testing of large numbers of additional TALE drivers, a task that exceeds the scope of the current proof of concept manuscript. Finally, based on the reviewer's suggestion, we have now revised the Discussion section to more clearly place the broad spectrum of possible applications in the context of future directions.

Figure R3: TALE off-target analysis. (a, b) Genome-wide *in silico* prediction of off-target sites containing up to four mismatches within a 4kb window (-2kb to + 2kb) centred on the transcription start sites (TSS) of annotated *Drosophila* genes. Bar graph (a) and matrix (b) show the number of affected genes relative to the number of mismatches for each of the four TALEs. **(c)** Genomic locations of candidate off-target loci for TALE₃ (CG16890) and TALE₄ (CG5613) (diagrams adapted from FlyBase:GBrowse). The number of base pairs reflects the distance between the off-target site and the TSS of predicted target genes. All off-target loci contained four nucleotide mismatches relative to the corresponding on-target sequence. **(d)** Quantification of relative expression (RT-qPCR) from the on-target transgenes and off-target endogenous genes in wild-type (w^{1118} , grey), triple TALE driver (yellow), triple VAS responder (light green), and triple TALE driver/triple VAS responder (dark green) third larval instars (n = 3 biological replicates (x3 technical replicates); mean +/- s.d.).

REFERENCES

1. zur Lage, P. & Jarman, A.P. Antagonism of EGFR and notch signalling in the reiterative recruitment of Drosophila adult chordotonal sense organ precursors. *Development* **126**, 3149-3157 (1999).
2. Zhu, S. et al. Gradients of the Drosophila Chinmo BTB-zinc finger protein govern neuronal temporal identity. *Cell* **127**, 409-422 (2006).
3. Jarman, A.P., Brand, M., Jan, L.Y. & Jan, Y.N. The regulation and function of the helix-loop-helix gene, *asense*, in Drosophila neural precursors. *Development* **119**, 19-29 (1993).
4. Pfeiffer, B.D. et al. Tools for neuroanatomy and neurogenetics in Drosophila. *Proc Natl Acad Sci U S A* **105**, 9715-9720 (2008).
5. Crocker, J. & Stern, D.L. TALE-mediated modulation of transcriptional enhancers in vivo. *Nat Methods* **10**, 762-767 (2013).
6. Pfeiffer, B.D., Truman, J.W. & Rubin, G.M. Using translational enhancers to increase transgene expression in Drosophila. *Proc Natl Acad Sci U S A* **109**, 6626-6631 (2012).

REVIEWERS' COMMENTS:

Reviewer #1 (Remarks to the Author):

The authors have provided in-depth, thoughtful responses to each of my comments, and have added very helpful new data and explanations to the manuscript. I believe it is now suitable for publication.

Reviewer #2 (Remarks to the Author):

I thank the authors for addressing my concerns. This is satisfactory.